# Probiotic Sheep Milk Ice Cream with Inulin and Apple Fiber

**DOI:** 10.3390/foods10030678

**Published:** 2021-03-22

**Authors:** Magdalena Kowalczyk, Agata Znamirowska, Magdalena Buniowska

**Affiliations:** Department of Dairy Technology, Institute of Food Technology and Nutrition, University of Rzeszow, Ćwiklińskiej 2D, 35601 Rzeszów, Poland; aznam@univ.rzeszow.pl (A.Z.); mbuniowska@ur.edu.pl (M.B.)

**Keywords:** ice cream, sheep’s milk, probiotics, apple fiber, inulin, *Bifidobacterium*, *Lactobacillus*

## Abstract

The aim of the study was to assess the effect of the addition of inulin and the replacement of part of the inulin with apple fiber on the physicochemical and organoleptic properties of ice cream. Moreover, the survival of *Bifidobacterium animalis* ssp. *Lactis* Bb-12 and *Lactobacillus rhamnosus* was studied in sheep milk ice cream. There was no effect of the apple fiber and the type of bacteria on the number of bacteria in the probiotics after fermentation. As a result of freezing, in the mixture containing *Bifidobacterium animalis* ssp. *Lactis* Bb-12, there was a significant reduction in the bacteria from 0.39 log cfu g^−1^ to 0.46 log cfu g^−1^. In all of the ice cream on the 21st day of storage, it exceeded 10 log cfu g^−1^, which means that the ice cream retained the status of a probiotic product. The *Lactobacillus rhamnosus* ice cream showed a lower yellow color compared to the *Bifidobacterium* Bb-12 ice cream. The overrun of the sheep’s milk ice cream was within the range of 78.50% to 80.41%. The appearance of the sheep’s milk ice cream is influenced significantly by the addition of fiber and the type of bacteria and the interaction between the type of bacteria and the addition of fiber, and storage time and fiber.

## 1. Introduction

The growing awareness of consumers and their expectations regarding healthy and good quality food has contributed to increased demand for functional food production. Bioactive ingredients added to food positively affect the product’s characteristics and quality, and positively affects human health [1]. The trend of using sheep’s milk for ice cream production has recently emerged. Compared to other mammals’ milk, sheep’s milk has a higher nutritional value, with much more dry matter. Sheep’s milk is a source of essential minerals and vitamins for the body [2].

The popularity of ice cream consumption and its availability contributed to developing functional ice cream recipes with increased nutritional value, enriched with probiotic bacteria and prebiotics [3]. According to FAO/WHO (Food and Agriculture Organization of the United Nations and the World Health Organisation) probiotic bacteria are live microorganisms that provide health benefits to the host [4]. However, to maintain the probiotic effect, a minimum number of viable probiotic cells of 10^6^–10^9^ CFU (colon forming unit) is required [5].

*Bifidobacterium animalis* ssp. *Lactis* Bb-12 (*Bifidobacterium* Bb-12) and *Lactobacillus rhamnosus* (*Lb. rhamnosus*) are among the most commonly used probiotics from the group of lactic acid bacteria. Probiotics have antioxidant, anti-inflammatory, antibacterial, and antiviral effects [6]. Moreover, probiotics have beneficial effects on different immune disorders, encompassing rheumatoid arthritis, inflammatory bowel disease (IBD) [7,8]. However, dietary fiber intake is still low [9,10]. In such a case, synbiotic ice cream can be a proposition for supplementing fiber in the diet. Numerous studies show that prebiotics support beneficial health effects, including stimulating the absorption of minerals (especially iron and calcium [11]), accelerating fat metabolism, facilitating the treatment of obesity, and preventing constipation [12]. Clinical studies show that natural polysaccharides [13,14,15] affect mainly the growth and survival of the bacteria *Lactobacillus* and *Bifidobacterium* among others including inulin, pectin, galactooligosaccharides (GOS), fructooligosaccharides (FOS) [16]. Inulin can be used in ice cream as a replacer for fat or sugar. Inulin plays a technological role by limiting ice crystal growth during freezing and storage, changing the mixture’s freezing point, and influencing ice cream’s melting. Apple fiber [17], obtained from the cleaning, micronization, and sterilization of dry apple pomace, also has a prebiotic potential. Apple fiber is a source of water-soluble pectin that is not digested by enzymes in the human digestive system [18]. After consumption, apple fiber reaches the small and large intestines relatively unchanged, nourishing the colonizing probiotics. Therefore, the study aimed to evaluate the effects of inulin addition and replacement of inulin with apple fiber on the physicochemical and organoleptic properties and the survival of *Bifidobacterium* Bb 12 and *Lb. rhamnosus* in sheep milk ice cream.

## 2. Materials and Methods

### 2.1. Materials

The material for the production of the ice cream was raw sheep’s milk (Farm “Owcza Zagroda,” Wyżne, Poland), with the following chemical composition: 5.34 ± 0.2% protein, 6.20 ± 0.3% fat, 5. 01 ± 0.12% lactose determined on a Bentley milk and milk product analyzer (Bentley, Minneapolis, MN, USA) and pH 6.8 ± 0.12 (FiveEasy pH-meter, Mettler Toledo, Greifensee, Switzerland).

For the production of the ice cream, the following were used: white sugar (Polish sugar, Rzeszów, Poland), mango–passion fruit flavor essence (Brown, Poland) with the following composition: natural and identical to raw mango and mango–passion essence, citric acid E330 and mango juice, inulin (Orafti HP, Oreye, Belgium), apple fiber (Aura Herbals Jarosław Paul, Sopot, Poland) composed of 100% micronized apple fiber. To inoculate the sample, probiotic bacteria (Chr. Hansen, Hoersholm Denmark) were used: *Bifidobacterium* Bb-12 and *Lb. rhamnosus* (Pen, Oxy).

### 2.2. Manufacture of Ice Cream Mixtures

Sheep’s milk (85%), sugar (11%), and mango–passion essence (0.1%) were mixed and divided into two batches. Inulin (4%) was added to the first batch and divided into a sample containing *Bifidobacterium* Bb-12 (Abb12: sample with 4% inulin and *Bifidobacterium* bb12) and *Lb. rhamnosus* (BLr—sample with 4% inulin and *Lb. rhamnosus*,). Inulin (2.5%) and apple fiber (15%) were added to the second batch, and they were also divided into two groups, AFbb12 (sample with 2.5% inulin, 1.5% apple fiber and *Bifidobacterium* bb12) and BFLr (sample with 2.5% inulin, 1.5% apple fiber and *Lb. rhamnosus).* The milk with the additives was mixed and homogenized using a homogenizer (Nuoni GJJ-0.06/40, Zhejiang, China) with a pressure of 20 MPa at 60 °C. After, the milk was pasteurized at 85 °C for 30 min and cooled to 37 °C. The groups Abb12 and AFbb12 were inoculated with a monoculture of *Bifidobacterium* Bb-12, while the groups Blr and BFLr were inoculated with *Lb. rhamnosus*. Each prepared milk sample was inoculated with a previously activated starter culture according to the method of Mituniewicz–Małek et al. [19] with some modification. After 5 h, the inoculum consisted of log 9 cfu g^−1^ of bacteria, which was added to the milk in the amount of 5%. Prepared mixtures were fermented in an incubator (Cooled Incubator ILW 115, POL-EKO-Aparatura, Wodzisław Śląski, Poland) at 37 °C for 10 h, then cooled to 5 °C and conditioned at this temperature for 12 h. Prepared ice cream mixes were frozen in a freezer (UNOLD AG, Hockeheim Germany) for 40–50 min. The produced ice cream was packed in 100 mL plastic containers and stored at −22 °C for twenty-one days.

### 2.3. Physicochemical Analysis

The chemical composition of ice cream and ice cream mixes were determined using a Bentley B-150 milk and dairy analyzer (Bentley, Minneapolis, MN, USA). Measurement of the pH value of the milk, ice cream mixes, and ice cream was performed with a FiveEasy pH meter (Mettler Toledo, Greifensee, Switzerland). Lactic acid content was determined by titration of the samples with 0.1M NaOH using phenolphthalein as an indicator. The results are expressed in g L^−1^ [20]. Ice cream overrun was estimated as the air volume ratio in the ice cream to the melted ice cream volume [21]. The melting and the first dropping time were assessed at an ambient temperature of 23 °C by placing a defined ice cream sample on stainless steel grids.

### 2.4. Microbiological Analysis

The number of probiotic bacteria *Bifidobacterium* Bb-12 and *Lb. rhamosus* were determined according to the method of Lima et al. [22]. The inoculation was done by the plate method using MRS agar (Biocorp, Warszawa, Poland) and then incubated anaerobically with GENbox anaer (Biomerieux, Warszawa, Poland) in a vacuum desiccator at 37 °C for 72 h [23]. The colonies were counted using a colony counter (TYPE J-3, Chemland, Stagard Szczeciński, Poland). The number of viable bacterial cells was expressed as log cfu g^−1^.

### 2.5. Color of Ice Cream

The color of the ice cream was determined by the instrumental method using a colorimeter (model NR 145, Shenzhen, China) using the CIE LAB system [24]. The following values were analyzed: L* as brightness, and as a* color from red (+) to green (−), b* as the colors from yellow (+) to blue (−), C* as the purity and intensity of the color, and h* as the shade of the color.

### 2.6. Organoleptic Analysis

The organoleptic evaluation was carried out by 20-person panel experts. The parameters were assessed on a 9-degree scale with structures and definitions.

### 2.7. Statistical Analysis

The results from two independent studies were expressed as the mean and standard deviation in Statistica v. 13.1 (StatSoft, Tulsa, OK, USA). One, two, and three-way ANOVA was performed, and the differences between the mean values were verified with the Turkey test, with *p* < 0.05.

## 3. Results

### 3.1. Physicochemical and Organoleptic Properties of Ice Creams Mixtures and Ice Creams

The chemical composition of ice cream mixes based on sheep’s milk before and after fermentation is presented in Table 1. There was no significant effect of fermentation and of the apple fiber on the protein and fat concentration in the ice cream mixes. The protein content of all ice cream mixes was around 5%, while the fat content ranged from 5.9% to 6.1%. In the studies by Góral et al. [25], probiotic milk ice cream contained 6.9% to 7.5% protein and 5.1% to 5.6% fat. A similar fat content (5.77–5.90%) in milk ice cream with strawberries and probiotic bacteria was shown by Vardar and Öksüz [26]. Akalin and Erişir [27] prepared a probiotic milk ice cream with a fiber content of 4%, and total solids at 33%. Homayouni et al. [28] showed a higher content of dry matter at 38.5% and fat at 8.1% in synbiotic ice cream. On the other hand, Balthazar et al. [2] prepared ice cream made of sheep’s milk with a higher fat content (10.03%) and lower protein (3.2%).

The carbohydrate content of ice cream blends before fermentation ranged from 19.45% in AFbb12 to 19.50% in Abb12 and was not significant. As expected, the carbohydrate content after fermentation decreased by 1.3–1.4%. There was no significant effect of the apple fiber and the type of bacteria used for fermentation on the carbohydrate content in sheep’s milk ice cream mixes. Also, in the study by Góral et al. [25], no significant effect of additives on the carbohydrate content (26.52–27.48%) in the ice cream was found. In sheep’s milk ice cream, in a study by Balthazar et al. [2], the carbohydrate content was determined in the range from 18.1 to 18.6%, i.e., similar to the results presented in Table 1 with the chemical composition of ice cream such as protein, fat and carbohydrates. The effect of storage time, the addition of apple fiber, and the type of fermenting bacteria on the pH value and the concentration of lactic acid in sheep’s milk ice cream are presented in Table 2. 

Ice cream mixes with apple fiber (AFbb12 and BFLr) were characterized by a significantly lower pH value than the mixes made only with inulin (*p* ≤ 0.05). 

In mixtures containing *Bifidobacterium* Bb-12, a lower pH value was significant in the mixtures with *Lb. rhamnosus.* Ice cream mixtures also showed a lower lactic acid content of 0.23 g L^−1^ and 0.29 g L^−1^ compared to Abb12 and AFbb12 blends. The conducted three-factor ANOVA (Table 3) shows that the pH value is significantly influenced by the three analyzed research factors (type of bacteria, storage time, apple fiber) and the interactions of these factors. The effect of storage time on the pH value of the ice cream mixes and ice cream is mainly due to the inclusion of the pH value before fermentation in this comparison. There was no significant effect of the storage time on the mixtures’ pH values after fermentation and on the ice cream after 7 and 21 days of storage. The addition of 1.5% apple fiber resulted in maintaining lower pH values in the mixes and ice cream throughout the entire study period. In the study carried out by Akalin and Erisir [27], the pH value in the range of 5.35 to 5.45 was determined in probiotic ice cream with the addition of oligofructose and inulin. In milk ice cream containing *Lb. rhamnosus*, Pankiewicz et al. [29] determined the pH value as 5.73 to 5.83. 

### 3.2. Microbiological Analysis of Ice Cream Mixtures and Ice Creams

The presented pH values of the product help to maintain the high survival rate of probiotic bacteria (Table 2 and Table 4). In addition, some studies confirmed a higher pH in fermented ice cream than in fermented milk or in fermented frozen desserts [30].

Mohammadi et al. [31] and Da Silva et al. [32] reported the pH value of 6.45 in unfermented ice cream with the addition of *Bifidobacterium* Bb-12. In addition, some studies have found a higher pH in fermented ice cream than in fermented milk or in fermented frozen desserts. Ozturk et al. [33] determined the pH value in fermented ice cream from 5.28 to 5.89, depending on the additives used. These low pH values determined by Dos Santos et al. [34] and Ozturk et al. [33] were associated with the addition of fruits, which lower the pH value. 

The ice cream with the addition of AFbb12 and BFLr apple fiber also showed a higher content of lactic acid after 7 and 21 days than ice creams with inulin addition. The ANOVA analysis of variance indicates that the concentration of lactic acid was significantly influenced by the type of bacteria, the addition of fiber, and the interaction between the type of bacteria and fiber. Akalin et al. [35] reported that the presence of various dietary fibers influences the lactic acid content, especially in ice cream with orange, apple, and bamboo fiber. Those authors, in probiotic ice cream with 2% apple fiber, added as much as 3.65 g/100 g of lactic acid.

As a result of the ten-hour fermentation of the ice cream mixes from sheep’s milk by *Lb. rhamnosus* and *Bifidobacterium* Bb-12, the number of bacterial cells exceeded 11 log cfu g^−1^ (Table 4).

In the process of ice cream manufacturing, the ingredients used in the recipe may adversely affect the probiotic by changing the pH (e.g., pH 5.5–6.0 is optimal for the growth of *Lactobacillus acidophilus* and pH 6.0–7.0 is favorable for *Bifidobacterium*), titratable acidity or sugar content [31,36]. In this case, there was no effect from the addition of apple fiber and the type of bacteria on the number of viable cells after fermentation.

When the temperature is decreasing during the freezing of the ice mixtures, changes in the osmotic pressure in the cells result in changes in the microorganisms, causing the loss of their metabolic properties. During the freezing process, the formed ice crystals can mechanically damage cell walls, and the condensation of harmful solutes or dehydration of cells additionally intensify the adverse changes [37,38]. The adverse effect of oxygen due to the aeration process during freezing and high redox potential values on anaerobic bacteria, especially *Bifidobacterium* [31,36], should also be mentioned. The survival rate of probiotic bacteria depends on the bacteria, production technology, temperature, storage time, and ice cream chemical composition. The results presented in Table 4 indicate that as a result of freezing ice mixes fermented by *Bifidobacterium* Bb-12, there was a significant reduction in the bacterial number from 0.39 log cfu g^−1^ to 0.46 log cfu g^−1^ compared to the number of cells of these bacteria in the mixtures after fermentation (*p* ≤ 0.05). The low pH of the mixtures and the high content of lactic acid contributed to reducing the *Bifidobacterium* Bb-12. The decrease in bacterial cell counts resulting from freezing was likely due to damage to the bacterial cell walls that led to the bacterial cells’ death [39]. In the studies of Akalin and Erisir [27], during the freezing of mixtures with *Lactobacillus acidophilus* and *Bifidobacterium* Bb-12, the number of bacterial cells decreased by 1.5- to 2 log CFU. Table 4 shows no significant effect of the freezing process on the number of *Lb. rhamnosus* cells in BLr and BFLr ice cream. The lack of this effect on the number of *Lb. rhamnosus* cells can be explained by significantly higher pH values and lower lactic acid content. According to Godward et al. [40] and Tamime et al. [41], probiotic bacteria’s resistance to pH and acidity is bacteria dependent. It was found that *Lactobacillus* has a broad cytoplasmic buffering capacity and resistance to pH (3.72–7.74), which enables its stability and resistance to changes in cytoplasmic pH in an acidic environment.

Also, the 1.5% addition of apple fiber did not significantly affect the number of viable bacterial cells immediately after freezing (Table 4). Mohammadi et al. [31] obtained 8 log CFU mL^−1^ of *Lactobacillus acidophilus* and 8 log CFU mL^−1^ of *Bifidobacterium bifidum* immediately after freezing the ice cream. Akbari et al. [42] reported that after freezing, the viability of bacteria decreased by 0.28 (*Lactobacillus acidophilus*) and by 0.33 (*Lb. rhamnosus*) log CFU.

In these studies, ice cream storage at −22 °C for 7 and 21 days resulted in insignificant reduction of both *Bifidobacterium* Bb-12 and *Lb. rhamnosus* (Table 4). However, after 7 and 21 days of storage, many bacterial cells were determined in ice cream with *Lb. rhamnosus* BLr and BFLr. The number of viable *Lb. rhamnosus* and *Bifidobacterium* Bb-12 cells in all of the ice cream on day 21 of storage exceeded 10 log cfu g^−1^, which means that the ice cream maintained its probiotic status (Table 4). The probiotic ice cream tested by Góral et al. [25] also showed a high number of bacterial cells in the range from 9 log CFU mL^−1^ to 11 log CFU mL^−1^. On the other hand, in the studies by Akalin and Erisir [27], a decrease in the bacterial number (from 0.3 to 0.9 log cfu g^−1^) was found during ice cream storage. According to the International Dairy Federation’s recommendations, products defined as probiotic should contain at least 7 log cfu g^−1^
*lactobacillus* or 6 log cfu g^−1^
*Bifidobacterium* [43]. The studies of Balthazar [44] showed the number of *Lactobacillus acidophilus* cells exceeding 6 log cfu g^−1^ in probiotic ice cream. Similarly, Akalin and Erisir [27] reported that probiotic cultures had an excellent ability to survive and maintain high cell counts in frozen foods.

The excellent survival rate of probiotic bacteria cells obtained in these studies ensures that the therapeutic level of synbiotic sheep’s milk ice cream is maintained for at least 21 days. According to Jayamann and Adams [45], a bacterial level of 7 log cfu g^−1^ is required to obtain a therapeutic (anti-diarrheal) effect.

### 3.3. Color of Ice Cream

The results of the color of the ice cream mixes and ice cream during storage are presented in Table 5.

The L* parameter’s highest values were recorded in the ice cream mix with inulin and *Lb. rhamnosus*, and then in the mix with inulin and *Bifidobacterium* Bb-12. The lighter color of the Abb12 and BLr ice cream was maintained throughout the storage period. The 1.5% addition of apple fiber decreased the color brightness by about 9 units in the AFbb12 and BFLr blends. After 7 and 21 days of storage, an increase in L* brightness was found in all ice cream samples. Extending the storage time from 7 to 21 days resulted in a further increase in L* brightness, but the differences were not significant. A significant effect of the storage time, the addition of apple fiber and the interaction of these two factors on the brightness of the color of the ice cream was demonstrated (Table 3).

Ice cream mixes and ice cream with the addition of apple fiber (AFbb12 and BFLr) were characterized by a higher red color (+a*), which comes from the phenolic compounds and pectin contained in the fiber [45,46]. On the other hand, only ice cream with inulin addition had a higher green color parameter (a*). The ANOVA analysis of variance shows that a* parameter was influenced by the type of bacteria, storage time, the addition of apple fiber, and interactions between the storage time and the addition of apple fiber.

The analyzed ice cream and ice cream mixture were characterized by a high value of yellow (+b*) due to the mango–passion fruit essence used in their production. A significantly lower value of yellow color was found in the ice cream fermented with *Lb. rhamnosus* throughout the storage period (*p* ≤ 0.05). These results were confirmed by the analysis of variance, which showed that the type of bacteria, the addition of fiber, and the interaction of these two factors significantly influenced the yellow color intensity. The results for the intensity (C*) and hue (h*) of the color indicate that these color coordinates are dependent on all of the factors tested (type of bacteria, storage time, and fiber addition) and their interactions. The addition of apple fiber decreased the h* color saturation parameter in AFbb12 and BFLr ice cream. The studies of Akalin et al. [35] also found that the addition of orange and apple fiber reduces the L* brightness of ice cream and intensifies the red and yellow color. Also, in the studies of Crizel et al. [47], Dervisoglu and Yazici [48], ice cream with citrus fiber added lower L* values and higher a* and b * values, which indicates that the addition of fiber causes a reduction in brightness and gives a red and yellow color. Ice cream made by Calligaris et al. [49] determined the following values of the color parameters: L* 87.93, a* 2.41, and b* 6.91. In the studies carried out of Pankiewicz et al. [29], L* brightness in the range from 71.15 to 71.31 and the a* 0.15 and b* 13.29–13.64 color coordinates were determined in milk ice cream fermented by *Lb. rhamnosus*. Table 6 shows the analysis of sheep’s milk ice cream’s physical properties for such features as overrun, first dropping time, and melting time. One of the most important parameters in assessing the quality of ice cream is the degree of air entrainment in the ice cream mixture, i.e., overrun. The ice cream mixture ingredients have the most significant impact on overrun, especially the content and protein proportion to fat [50].

The results in Table 6 show that the ice cream’s overrun was not affected by the storage time, the addition of apple fiber, and the type of bacteria used to ferment the mixture. The sheep’s milk ice cream overrun was from 78.50% to 80.41% (Table 6). The studies of Akalin et al. [35] showed a lower overrun of ice cream from 25.55% to 30.60%; the 2% proved that the addition of apple fiber increased the overrun of ice cream compared to ice cream without this addition. On the other hand, Akin et al. [38] indicated that the ice cream’s overrun depended on the content of sugar and inulin. Increasing the sugar content from 15% to 21% resulted in an increased overrun from 27.8% to 32.3%. However, Crizel et al. [47] proved that the addition of orange fiber as a fat replacement resulted in a significant reduction in the ice cream’s overrun compared to the control sample, probably due to the lower fat content. In the conducted research, all ice cream groups contained about 6% fat; hence their overruns did not differ. Our study also proves that replacing inulin with apple fiber does not change the overrun of ice cream.

The melting rate of ice cream is influenced by many factors, including the total dry matter content, ice crystals, size, and the number of fat globules [25]. The first dropping time of ice cream drip time after 7 days of freezer storage differed significantly depending on the type of bacteria used to ferment the mixture. Ice creams containing *Lb. rhamnosus* (BLr and BFLr) had a longer time for the first dropping time after 7 days of storage than those containing *Bifidobacterium* Bb-12 (Table 6). Extending the ice cream storage time to 21 days significantly reduced the first dropping time by 19–22 s in all ice cream groups.

A fast-melting product is undesirable, and a too slow melting rate can also be a disadvantage of ice cream [51]. The total melting time was shorter in the apple fiber ice cream (AFbb12 and BFLr) than their inulin only counterparts (Abb12 and BLr). Zhang et al. [51] found that the addition of pectin reduced ice cream’s melting rate and led to a more excellent product stabilization. The pectin present in apple fiber may interact with other milk components to create a dense three-dimensional network structure and reduce the heat transfer rate. [52] reported that apple fiber’s addition increased the freezing temperature and led to a decrease in ice crystals and the percentage of frozen water. The analysis of variance carried out indicates that the total melting time was influenced by the interactions between the storage time and the addition of apple fiber and the interactions between the three examined factors (type of bacteria, storage time, and fiber). According to Criscio et al. [53] and El-Nagar et al. [54] samples containing 5% inulin had a significantly higher melting rate than the controls and samples with 2.5% inulin. In the studies of Balthazar et al. [44], probiotic ice cream with a 10% addition of inulin also had a longer melting time than the probiotic ice cream without inulin. Akin et al. [38], in their study on ice cream depending on inulin, noted that the values of the first dropping time and total melting time were within the ranges of 1780 s (15% sugar without inulin)–2058 s (21% sugar with 2% inulin), respectively, and 4806 s (21% sugar without inulin)–5313 s (18% sugar with 2% inulin). In other yogurt ice cream studies, the addition of 5% inulin reduced the melting rate from 5% to 9% due to binding water, thus reducing the interaction of dry matter components with water [54]. According to our research (Table 1), all ice cream groups did not differ in fat content. Therefore, no differences in overrun were found. In this case, the addition of inulin had a more significant effect on extending the ice cream melting time. Analyses show that ice cream with 2.5% inulin and 1.5% apple fiber (AFbb12, BFLr) is quicker to melt than ice cream with 4% inulin (Abb12, BLr). Also, in the studies of Akalin and Erisir [27], the addition of inulin increased the first dropping time and the total time of ice cream melting.

From a technological point of view, the use of fruit fiber in ice cream production causes significant changes in organoleptic characteristics, improves their taste and texture [55].

The addition of apple fiber significantly changed the appearance of sheep’s milk ice cream. It increased sandiness and intensified the additives taste, and the smell of the expansion of mango–passion fruit essence, both on the 7th and 21st day of freezing storage (Table 7).

Akalin et al. [35] showed that, in the ice cream samples prepared with apple and orange fiber, lower scores were given for flavor characteristics compared to the control ice cream. Crizel et al. [47] also showed lower taste scores for ice cream with 1.5% orange fiber than for the controls. The addition of microorganisms and inulin in the studies of Criscio et al. [53] did not significantly affect flavor intensity, texture, and smoothness. On the other hand, Akbari et al. [42] indicated that the introduction of inulin as a fat substitute made the evaluators prefer the taste of the ice cream containing 3% and 4% of inulin to the taste of the ice cream without inulin. Akin et al. [38] conducted studies on inulin and sugar influence on prebiotic ice cream’s physical and sensory properties. They found that the addition of inulin does not affect the sensory properties of ice cream. The analysis of variance performed shows that the storage time, type of bacteria, and interactions of all examined factors (storage time, kind of bacteria, fiber) do not affect the hardness, smoothness, sweet taste, taste, and smell of additives as well as foreign taste and smell. The research indicates that only apple fiber’s addition contributed to a change in the smoothness additives, taste and smell.

## 4. Conclusions

Sheep’s milk ice cream can be a good source of probiotic bacteria and dietary fiber. The addition of 1.5% apple fiber instead of inulin resulted in a change in the ice cream’s physicochemical and organoleptic characteristics. On an industrial scale, when using the addition of apple fiber for the production of probiotic ice cream, attention should be paid to selecting the strain and the survival of probiotic bacteria during the freezing of the mixture and storage of ice cream.

## Figures and Tables

**Table 1 foods-10-00678-t001:** Chemical composition of ice cream mixture samples.

Chemical Composition	Storage Time (Days)	Abb12	AFbb12	BLr	BFLr
Protein [%]	0	4.97 ^Aa^ ± 0.04	4.98 ^Aa^ ± 0.10	4.91 ^Aa^ ± 0.02	4.95 ^Aa^ 5 ± 0.04
	1	4.98 ^Aa^ ± 0.11	5.0 ^Aa^ ± 0.05	4.90 ^Aa^ ± 0.17	4.92 ^Aa^ ± 0.05
Fat [%]	0	6.08 ^Aa^ ± 0.23	6.00 ^A^ ± 0.04	6.07 ^Aa^ ± 0.02	6.04 ^Aa^ ± 0.03
	1	6.10 ^Aa^ ± 0.22	5.97 ^Aa^ ± 0.03	6.05 ^Aa^ ± 0.20	6.03 ^Aa^ ± 0.02
Carbohydrates [%]	0	19.50 ^Ba^ ± 0.14	19.45 ^Ba^ ± 0.02	19.49 ^Ba^ ± 0.04	19.46 ^Ba^ ± 0.02
	1	18.04 ^Aa^ ± 0.06	18.07 ^Aa^ ± 0.09	18.10 ^Aa^ ± 0.03	18.14 ^Aa^ ± 0.07

Mean ± standard deviation. *n* = 20; ^a^—Mean values denoted in rows by different letters differ statistically significantly at (*p* ≤ 0.05); ^A,B^—Mean values in columns obtained for a given parameter denoted by different letters differ significantly (*p* ≤ 0.05). Abb12: sample with 4% inulin and *Bifidobacterium* bb12, AFbb12: sample with 2.5% inulin, 1.5% apple fiber and *Bifidobacterium* bb12, BLr: sample with 4% inulin and *Lb. rhamnosus*, BFLr: sample with 2.5% inulin, 1.5% apple fiber and *Lb. rhamnosus*. Time: 0 before fermentation, 1 after fermentation.

**Table 2 foods-10-00678-t002:** Lactic acid content and pH value of ice creams during storage.

Properties	Storage Time (Days)	Abb12	AFbb12	BLr	BFLr
pH	0	6.60 ^Bb^ ± 0.03	6.24 ^Ba^ ± 0.01	6.61 ^Bb^ ± 0.01	6.24 ^Ba^ ± 0.01
	1	5.19 ^Ab^ ± 0.08	4.93 ^Aa^ ± 0.03	5.97 ^Ad^ ± 0.04	5.75 ^Ac^ ± 0.03
	7	5.16 ^Ab^ ± 0.05	4.90 ^Aa^ ± 0.03	5.94 ^Ad^ ± 0.02	5.72 ^Ac^ ± 0.02
	21	5.20 ^Ab^ ± 0.08	4.90 ^Aa^ ± 0.03	5.91 ^Ad^ ± 0.04	5.71 ^Ac^ ± 0.02
Lactic acid [g/L]	1	0.61 ^Ac^ ± 0.04	0.71 ^Ad^ ± 0.04	0.38 ^Aa^ ± 0.01	0.42 ^Ab^ ± 0.01
	7	0.62 ^Ab^ ± 0.08	0.74 ^Ac^ ± 0.06	0.39 ^Aa^ ± 0.05	0.41 ^Aa^ ± 0.02
	21	0.62 ^Ab^ ± 0.01	0.75 ^Ac^ ± 0.03	0.38 ^Aa^ ± 0.01	0.40 ^Aa^ ± 0.02

Mean ± standard deviation. *n* = 20; ^a–d^—Mean values denoted in rows by different letters differ statistically significantly at (*p* ≤ 0.05); ^A,B^—Mean values in columns obtained for a given parameter denoted by different letters differ significantly (*p* ≤ 0.05). Abb12: sample with 4% inulin and *Bifidobacterium* bb12, AFbb12: sample with 2.5% inulin, 1.5% apple fiber and *Bifidobacterium* bb12, BLr: sample with 4% inulin and *Lb. rhamnosus*, BFLr: sample with 2.5% inulin, 1.5% apple fiber and *Lb. rhamnosus*. Storage time: 0 before fermentation, 1 after fermentation, 7 after 7 days, 21 after 21 days.

**Table 3 foods-10-00678-t003:** Analysis of variance (ANOVA) *p*-values on the effects of storage time and type of bacteria and fiber on color, pH, lactic acid, overrun, bacteria appearance, hardness, smoothness, sweet taste, additives taste, off taste, odor additives, and off odor of ice cream.

Properties	Type of Bacteria *p*-Values	Storage Time (Days) *p*-Values	Fiber *p*-Values	Type of Bacteria * Storage Time *p*-Values	Type of Bacteria * Fiber *p*-Values	Storage Time * Fiber *p*-Values	Type of Bacteria * Storage Time * Fiber *p*-Values
L*	n.s. 0.1813	↑ 0.0007	↑ 0.0000	n.s. 0.4748	n.s. 0.9892	↑ 0.0012	n.s. 0.2122
a*	↑0.0037	↑ 0.0073	↑ 0.0000	n.s. 0.8549	n.s. 0.1225	↑ 0.0072	n.s. 0.6325
b*	↑0.0000	n.s. 0.2994	↑ 0.0496	↑ 0.0213	↑ 0.0004	n.s. 0.1295	n.s. 0.5593
C*	↑0.0000	↑ 0.0228	↑ 0.0128	↑ 0.0000	↑ 0.0000	↑ 0.0048	↑ 0.0145
h*	↑0.0000	↑ 0.0083	↑ 0.0000	↑ 0.0421	↑ 0.0000	↑ 0.0000	↑ 0.0279
pH	↑0.0000	↑ 0.0482	↑ 0.0000	↑ 0.0258	↑ 0.0350	↑ 0.0426	↑ 0.0498
Lactic acid [g L^−1^]	↑0.0000	n.s. 0.3087	↑ 0.0000	↑ 0.0418	↑ 0.0323	n.s. 0.3110	n.s. 0.2388
Overrun [%]	n.s. 0.4132	n.s. 0.0786	n.s. 0.0786	n.s. 0.3532	n.s. 0.1096	n.s. 0.0701	n.s. 0.6300
First drop [s]	↑ 0.0012	↑ 0.0000	n.s. 0.0541	↑ 0.0001	n.s. 0.0531	n.s. 0.0620	n.s. 0.0714
Complete melting times [s]	↑ 0.0011	↑ 0.0000	↑ 0.0004	n.s. 0.0678	n.s. 0.0882	↑ 0.0000	↑ 0.0412
Bacteria	↑ 0.0096	↑ 0.0264	↑ 0.0390	↑ 0.0499	↑ 0. 0402	n.s. 0.2160	n.s. 0.1183
Appearance	↑0.0088	n.s. 0.3556	↑ 0.0158	n.s. 0.8324	↑ 0. 0426	↑ 0.01808	n.s. 0.2138
Hardness	n.s. 0.7617	n.s. 0.0870	n.s. 0.4301	n.s. 0.6735	n.s. 0.9116	n.s. 0.7844	n.s. 0.9940
Smoothness	n.s. 0.1067	n.s. 0.1559	↑ 0.0000	n.s. 0.1975	n.s. 0.1254	↑ 0.1860	n.s. 0.1103
Sweet taste	n.s. 0.4752	n.s. 0.3115	n.s. 0.5721	n.s. 0.8505	n.s 0.9699	n.s. 0.3590	n.s. 0. 5724
Additives taste	n.s. 0.5351	n.s 0.6157	↑ 0.0017	n.s. 0.7724	n.s 0.1524	n.s. 0.9151	n.s. 0.6372
Off taste	n.s. 0.1321	n.s. 0.9190	n.s. 0.9190	n.s. 0.9190	n.s. 0.9190	n.s. 0.1321	n.s. 0.1321
Odor additives	n.s. 0.2274	n.s. 0.3272	↑ 0.0022	n.s. 0.7721	n.s 0.0916	n.s. 0.9274	n.s. 0.7976
Off odor	n.s. 0.8243	n.s. 0.1248	n.s. 0.8243	n.s. 0.8243	n.s. 0.8243	n.s. 0.8243	n.s. 0.8243

* Storage time (days) = interaction ↑; Type of bacteria * Fiber = interaction ↑; Storage time * Fiber = interaction ↑; Type of bacteria * Storage time * fiber = interaction ↑; indicates significant effect *p* < 0.05; n.s. no significant effect.

**Table 4 foods-10-00678-t004:** Viable counts of probiotic bacteria in ice creams and ice creams mixture (log cfu g^−1^).

Storage Time (Days)	Abb12	AFbb12	BLr	BFLr
1	11.41 ^Ba^ ± 0.79	11.11 ^Ba^ ± 0.70	11.58 ^Aa^ ± 0.78	11.73 ^Aa^ ± 0.72
2	10.95 ^ABa^ ± 0.73	10.72 ^ABa^ ± 0.72	11.46 ^Ab^ ± 0.78	11.65 ^Ab^ ± 0.80
7	10.77 ^ABa^ ± 0.83	10.48 ^ABa^ ± 0.74	11.34 ^Ab^ ± 0.87	11.59 ^Ab^ ± 0.79
21	10.68 ^Aa^ ± 0.76	10.28 ^Aa^ ± 0.73	11.22 ^Ab^ ± 0.79	11.50 ^Ab^ ± 0.77

Mean ± standard deviation. *n* = 20; ^a,b^—Mean values denoted in rows by different letters differ statistically significantly at (*p* ≤ 0.05); ^A,B^—Mean values in columns obtained for a given parameter denoted by different letters differ significantly (*p* ≤ 0.05). Abb12: sample with 4% inulin and *Bifidobacterium* bb12, AFbb12: sample with 2.5% inulin, 1.5% apple fiber and *Bifidobacterium* bb12, BLr: sample with 4% inulin and *Lb. rhamnosus*, BFLr: sample with 2.5% inulin, 1.5% apple fiber and *Lb. rhamnosus*. Storage time: 1 after fermentation, 2 directly after freezing, 7 after 7 days, 21 after 21 days.

**Table 5 foods-10-00678-t005:** Color parameters of ice cream sample in the ice cream mixture during storage.

	Storage Time (Days)	Abb12	AFbb12	BLr	BFLr
L*	1	70.58 ^Ab^ ± 0.35	61.52 ^Aa^ ± 0.49	71.60 ^Ab^ ± 0.32	61.89 ^Aa^ ± 0.55
	7	85.14 ^Bb^ ± 2.35	74.22 ^Ba^ ± 1.86	88.30 ^Bb^ ± 2.46	75.97 ^Ba^ ± 0.43
	21	86.78 ^Bb^ ± 1.28	74.93 ^Ba^ ± 2.86	88.70 ^Bb^ ± 0.94	77.29 ^Ba^ ± 1.05
a*	1	−0.04 ^Bb^ ± 0.09	4.29 ^Ad^ ± 0.15	−0.75 ^Aa^ ± 0.04	3.27 ^Ac^ ± 0.16
	7	−0.37 ^Aa^ ± 0.18	5.23 ^Bb^ ± 0.67	−0.26 ^Ba^ ± 0.15	5.65 ^Bb^ ± 0.11
	21	−0.47 ^Aa^ ± 0.28	4.81 ^Bb^ ± 0.48	−0.39 ^Ba^ ± 0.10	5.30 ^Bb^ ± 0.29
b*	1	17.43 ^Ad^ ± 0.31	15.79 ^Ac^ ± 0.55	9.49 ^Aa^ ± 0.07	10.15 ^Ab^ ± 0.12
	7	20.07 ^Bc^ ± 3.11	17.63 ^Bc^ ± 2.85	10.46 ^Aa^ ± 0.73	12.64 ^Bb^ ± 1.03
	21	19.13 ^Bd^ ± 1.80	14.62 ^Ac^ ± 0.55	12.11 ^Ba^ ± 0.45	12.91 ^Bb^ ± 0.23
C*	1	17.43 ^Ad^ ± 0.31	16.16 ^Ac^ ± 0.45	9.52 ^Aa^ ± 0.08	10.66 ^Ab^ ± 0.14
	7	20.07 ^Bb^ ± 3.11	18.21 ^Bb^ ± 3.04	10.46 ^Aa^ ± 0.73	16.28 ^Cb^ ± 0.29
	21	19.14 ^Bc^ ± 1.81	15.46 ^Ab^ ± 0.54	12.12 ^Ba^ ± 0.46	13.77 ^Bab^ ± 0.59
h*	1	90.14 ^Ac^ ± 0.28	74.63 ^Bb^ ± 0.22	94.49 ^Bd^ ± 0.26	71.9 ^Ca^ ± 0.83
	7	90.74 ^Ac^ ± 0.92	73.41 ^ABb^ ± 1.36	90.82 ^Ac^ ± 0.71	69.52 ^Ba^ ± 0.57
	21	91.34 ^Ac^ ± 0.76	70.55 ^Ab^ ± 1.96	91.83 ^Ac^ ± 0.42	67.39 ^Aa^ ± 0.75

Mean ± standard deviation. *n* = 20; ^a–d^—Mean values denoted in rows by different letters differ statistically significantly at (*p* ≤ 0.05); ^A–C^—Mean values in columns obtained for a given parameter denoted by different letters differ significantly (*p* ≤ 0.05). Abb12: sample with 4% inulin and Bifidobacterium bb12, AFbb12: sample with 2.5% inulin, 1.5% apple fiber and *Bifidobacterium* bb12, BLr: sample with 4% inulin and *Lb. rhamnosus*, BFLr: sample with 2.5% inulin, 1.5% apple fiber and *Lb. rhamnosus*. Storage time: 1 after fermentation, 7 after 7 days, 21 after 21 days.

**Table 6 foods-10-00678-t006:** Overrun, first dropping time, and total melting rate in 7 and 21 days of storage.

Properties	Storage Time (Days)	Abb12	AFbb12	BLr	BFLr
Overrun [%]	7	79.15 ^Aa^ ± 0.20	80.41 ^Aa^ ± 0.22	80.50 ^Aa^ ± 0.95	78.50 ^Aa^ ± 0.12
	21	80.30 ^Aa^ ± 0.47	80.41 ^Aa^ ± 1.22	80.61 ^Aa^ ± 0.40	79.1 ^Aa^ ± 0.95
First drop [s]	7	972 ^Ba^ ± 12.21	960 ^Ba^ ± 10.26	991 ^Bb^ ± 8.71	982 ^Bb^ ± 10.12
	21	940 ^Ab^ ± 14.40	906 ^Aa^ ± 10.15	972 ^Ac^ ± 7.91	911 ^Aa^ ± 8.32
Complete melting times [s]	7	5469 ^Bb^ ± 35.20	5230 ^Ba^ ± 20.13	5913 ^Bd^ ± 38.12	5692 ^Bc^ ± 35.11
	21	4804 ^Ac^ ± 33,85	4007 ^Aa^ ± 25.42	5187 ^Ad^ ± 40.00	4201 ^Ab^ ± 15.32

Mean ± standard deviation. *n* = 20; ^a–d^—Mean values denoted in rows by different letters differ statistically significantly at (*p* ≤ 0.05); ^A,B^—Mean values in columns obtained for a given parameter denoted by different letters differ significantly (*p* ≤ 0.05). Abb12: sample with 4% inulin and *Bifidobacterium* bb12, AFbb12: sample with 2.5% inulin, 1.5% apple fiber and *Bifidobacterium* bb12, BLr: sample with 4% inulin and *Lb. rhamnosus,* BFLr: sample with 2.5% inulin, 1.5% apple fiber and *Lb. rhamnosus*. Storage time: 7 after 7 days, 21 after 21 days.

**Table 7 foods-10-00678-t007:** Sensory characteristics of ice cream on 7 and 21 days of storage.

Properties	Storage Time (Days)	Abb12	AFbb12	BLr	BFLr
Appearance	7	8.07 ^Ab^ ± 1.33	5.71 ^Aa^ ± 1.50	7.43 ^Ab^ ± 1.51	5.57 ^Aa^ ± 1.76
	21	8.25 ^Ab^ ± 1.39	5.75 ^Aa^ ± 0.50	8.75 ^Ab^ ± 0.50	5.00 ^Aa^ ± 1.15
Hardness	7	6.00 ^Aa^ ± 1.35	6.71 ^Aa^ ± 1.36	6.14 ^Aa^ ± 1.04	6.71 ^Aa^ ± 1.89
	21	6.38 ^Aa^ ± 1.20	6.75 ^Aa^ ± 1.26	6.00 ^Aa^ ± 0.82	6.85 ^Aa^ ± 1.26
Smoothness	7	7.07 ^Ab^ ± 1.77	3.14 ^Aa^ ± 1.91	6.86 ^Ab^ ± 1.07	3.00 ^Aa^ ± 1.31
	21	6.38 ^Ab^ ± 1.06	3.75 ^Aa^ ± 1.26	6.50 ^Ab^ ± 1.29	3.50 ^Aa^ ± 1.29
Sweet taste	7	5.43 ^Aa^ ± 1.03	4.43 ^Aa^ ± 0.53	5.24 ^Aa^ ± 1.07	4.57 ^Aa^ ± 1.62
	21	5.25 ^Aa^ ± 1.67	5.75 ^Aa^ ± 1.71	5.25 ^Aa^ ± 1.71	5.25 ^Aa^ ± 1.50
Additives taste	7	5.29 ^Aab^ ± 1.77	6.71 ^Ab^ ± 1.60	4.00 ^Aa^ ± 1.21	6.57 ^Ab^ ± 1.72
	21	5.75 ^Aab^ ± 1.05	6.50 ^Ab^ ± 1.29	4.25 ^Aa^ ± 1.71	7.25 ^Ab^ ± 1.50
Off taste	7	1.00 ^Aa^ ± 0.00	1.00 ^Aa^ ± 0.00	1.00 ^Aa^ ± 0.00	1.00 ^Aa^ ± 0.00
	21	1.00 ^Aa^ ± 0.00	1.00 ^Aa^ ± 0.00	1.00 ^Aa^ ± 0.00	1.00 ^Aa^ ± 0.00
Odor additives	7	3.00 ^Aa^ ± 1.71	4.00 ^Ab^ ± 1.00	1.71 ^Aa^ ± 0.76	4.29 ^Ab^ ± 1.56
	21	3.88 ^Aa^ ± 1.30	4.50 ^Ab^ ± 1.52	2.00 ^Aa^ ± 0.82	4.75 ^Ab^ ± 1.06
Off odor	7	1.00 ^Aa^ ± 0.00	1.00 ^Aa^ ± 0.00	1.10 ^Aa^ ± 0.00	1.00 ^Aa^ ± 0.00
	21	1.00 ^Aa^ ± 0.00	1.00 ^Aa^ ± 0.00	1.00 ^Aa^ ± 0.00	1.00 ^Aa^ ± 0.00

Mean ± standard deviation. *n* = 20; ^a,b^—Mean values denoted in rows by different letters differ statistically significantly at (*p* ≤ 0.05); ^A^ Mean values in columns obtained for a given parameter denoted by different letters differ significantly (*p* ≤ 0.05). Abb12: sample with 4% inulin and *Bifidobacterium* bb12, AFbb12: sample with 2.5% inulin, 1.5% apple fiber and *Bifidobacterium* bb12, BLr: sample with 4% inulin and *Lb. rhamnosus*, BFLr: sample with 2.5% inulin, 1.5% apple fiber and *Lb. rhamnosus*. Storage time: 7 after 7 days, 21 after 21 days.

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
