# Peer review of "Probiotic Sheep Milk Ice Cream with Inulin and Apple Fiber"

_foods, 2021, doi:10.3390/foods10030678_

Round 1

Reviewer 1 Report

The article “Probiotic sheep milk ice cream with inulin and apple fiber” aims to evaluate the effect of inulin addition and its partial replacement with apple fiber on the physicochemical and sensory properties of ice cream. The article is organized pretty well, however, as original article, I miss some interesting results.

In general, I suggest to short the discussion sections from lines 424 to 454 and from 477 to 493, they are too long.

L13: organoleptic properties of ice cream.

L17: in mixture …

L34: add a citation

L42: probiotic bacteria

L44-45: 106-109

L50-53: please revise this sentence

L63: I suggest “replacer” instead of “replacement”

L80: “5.34 ± 0.2% protein”, please change also fat and lactose

L101: groups

L102: replace “was” with “were”

L164: please correct the sentence

L166: ..

L186: indicate the unit of storage time, also in all other tables.

L190: 2.5% and 1.5%, same at line 191 and in all other captions

L204-205: the sentence is not clear, please rewrite it

L228: please correct the punctuation

L227: the sentence is not clear, please rewrite it

L285: log, please be coherent

L297: 8 log CFU mL-1

L310: the sentence is not clear, please rewrite it

L313: mL-1

L327: replace “presents” with “are presented in”

Author Response

Foods

Responses to comments

Title: Probiotic sheep milk ice cream with inulin and apple fiber

Authors: Magdalena Kowalczyk*, Agata Znamirowska and Magdalena Buniowska

Manuscript ID: foods-1131205

Responses to reviewers

To Reviewers 1:

General suggestion:

In general, I suggest to short the discussion sections from lines 424 to 454 and from 477 to 493, they are too long.

Response: Thank you for your suggestion. We have made corresponding modifications. We shorted these discussion sections

  1. Comment: Lines 13: organoleptic properties of ice cream.

Response: Thank you for your suggestion. We have made corresponding modifications.

  1. Comment: Lines 17: in mixture.

Response: Thank you for your suggestion. We have made corresponding modifications.

  1. Comment: Lines 34: add a citation.

Response:  citation is added

  1. Comment: Lines 42: probiotic bacteria.

Response: Thank you for your suggestion. We have made corresponding modifications

  1. Comment: Lines 44-45: 106-109.

Response: Thank you for your suggestion. We have made corresponding modifications

  1. Comment: Lines 50-53: please revise this sentence

Response: Thank you for your suggestion. We revised this sentence.

  1. Comment: Lines 63: I suggest “replacer” instead of “replacement”

Response: Thank you for your suggestion. We have made corresponding modifications

  1. Comment: Lines 80: “5.34 ± 0.2% protein”, please change also fat and lactose

Response: Thank you for your suggestion. We have made corresponding modifications

  1. 9. Comment: Lines 101: groups

Response: Thank you for your suggestion. We have made corresponding modifications

  1. Comment: Lines 102: replace “was” with “were”

Response: Thank you for your suggestion. We have made corresponding modifications

  1. Comment: Lines 164: please correct the sentence

Response: Thank you for your suggestion. We revised this sentence.

  1. Comment: Lines 166: ..

Response: Thank you for your suggestion. We revised this sentence.

 Comment: Lines 186: indicate the unit of storage time, also in all other tables.

Response:  Thank you for your suggestion. We added unit of storage time (days) in all tables

  1. Comment: Lines 190: 2.5% and 1.5%, same at line 191 and in all other captions

Response: Thank you for your suggestion. We have made corresponding modifications

  1. Comment: Lines 204-205: the sentence is not clear, please rewrite it

Response: Thank you for your suggestion. It was our mistake we decided to remove it

  1. Comment: Lines 228: please correct the punctuation

Response: Thank you for your suggestion. We corrected the punctuation

  1. Comment: Lines 227: the sentence is not clear, please rewrite it

Response: Thank you for your suggestion. We have made corresponding modifications. We revised this sentence.

  1. Comment: Lines 285: log, please be coherent

Response: Thank you for your suggestion. We have made corresponding modifications

  1. Comment: Lines 297: 8 log CFU mL-1

Response: Thank you for your suggestion. We have made corresponding modifications

  1. Comment: Lines 310: the sentence is not clear, please rewrite it

Response: We revised this sentence.

  1. Comment: Lines 313: mL-1

Response: Thank you for your suggestion. We have made corresponding modifications

  1. Comment: Lines 327: replace “presents” with “are presented in

Response: Thank you for your suggestion. We have made corresponding modifications

Reviewer 2 Report

This manuscript deals with an interesting and original study on the effect of inulin addition and replacement of inu-72 lin with apple fiber on the physicochemical and organoleptic proper-73 ties and the survival of Bifidobacterium Bb 12 and Lb. rhamnosus in sheep 74 milk ice cream.

The paper is clearly presented and results are very useful. However, I have some suggestions:

  1. Abstract, line 24, “considerably” is a not scientific word, please change it.
  2. Line 136: It will be interesting include total colour differences in section “2.5. Colour of ice cream” and discussion its results.
  3. Please, revise all table mean and standard deviation: the number of digits of the mean value depends on the place where the significant digit appears and the number of digits of the corresponding data should be adjusted by taking into account the corresponding standard deviation values. In this way, each value in the Tables has been expressed with the significant digits according to the significant digits of each standard deviation value. Correct the different errors in the tables, keeping in mind that 0 and 1 are not significant digits.

Author Response

To Reviewers 2:

This manuscript deals with an interesting and original study on the effect of inulin addition and replacement of inulin with apple fiber on the physicochemical and organoleptic properties and the survival of Bifidobacterium Bb 12 and Lb. rhamnosus in sheep milk ice cream.

  1. Comment: Abstract, line 24, “considerably” is a not scientific word, please change it.

Response: Thank you for your suggestion. We have made corresponding modifications and added word significantly

  1. Comment: Line 136: It will be interesting include total colour differences in section “2.5. Colour of ice cream” and discussion its results.

Response: Thank you for your suggestion. We have made corresponding modifications. Colour of ice cream is added in section 2.5 and 3.3.

  1. Comment: Please, revise all table mean and standard deviation: the number of digits of the mean value depends on the place where the significant digit appears and the number of digits of the corresponding data should be adjusted by taking into account the corresponding standard deviation values. In this way, each value in the Tables has been expressed with the significant digits according to the significant digits of each standard deviation value. Correct the different errors in the tables, keeping in mind that 0 and 1 are not significant digits.

Response: to show this table be more clearly, we added P-values in all parameters

Reviewer 3 Report

An interesting study on the use of different fiber qualities and effect op product quality. Comments are mostly on clarifying of the presentation.

Author Response

To Reviewers 3:

 1. Comment: Line 92: the flavor essence – if it is this the mango-passion essence, include this here

Response: Thank you for your suggestion. We have made corresponding modifications

  1. Comment: Line 121: give concentration of NaOH used

Response: Thank you for your suggestion. We have made corresponding modifications

 3. Comment: Line 164: …lower protein (3.2 something is missing here

Response: Thank you for your suggestion. We have made corresponding modyfications. 3.2 % was missing

 4. Comment: Line 166: remove point between …fat. Similar…

Response: Thank you for your suggestion. We have made corresponding modifications

 5. Comment: Line 170: content of 4 % and total solids 33 %

Response: Thank you for your suggestion. We have made corresponding modyfications

  1. Comment: Line 172: it says respectively, but what does the numbers refer to?

Response: Thank you for your suggestion. We have made corresponding modifications

  1. Comment: Line 174: is the numbers 19.45 % and 19.50 % significantly different (statistically)?

Response: Thank you for your suggestion. We have made corresponding modifications. Numbers was not significantly

  1. Comment: Line 186: indicate which chemical compositions that are presented in table 1

Response: Thank you for your suggestion. We have made corresponding modifications. We added indicated which chemical compositions: protein, fat and carbohydrates

  1. Comment: Line 203: as a result of the ten-hour fermentation – where is this presented? The table 2 only shows fermentation days 0, 1, 7 and 21

Response: Thank you for your suggestion. It was our mistake we decided to removed it.

  1. Comment: Line 210: pH value was significantly…

Response: Thank you for your suggestion. We have made corresponding modifications

  1. Comment: Line 211: research factors (type of bacteria…..

Response: Thank you for your suggestion. We have made corresponding modifications

  1. Comment: Line 227: give references for “some studies”

Response: Thank you for your suggestion. We have made corresponding modifications and added references

  1. Comment: Line 228: Mohammadi et al (35) and Da Silva…

Response: Thank you for your suggestion. We have made corresponding modifications

  1. Comment: Line 372: red and yellow color

Response: Thank you for your suggestion. We have made corresponding modifications

  1. Comment: Line 376: remove parentheses

Response: Thank you for your suggestion. We removed parentheses.

  1. Comment: Line 379: first dropping time comes out in Bold text – why?

Response: Thank you for your suggestion. We have made corresponding modifications

  1. Comment: Line 402: proved that the addition…..

Response: Thank you for your suggestion. We have made corresponding modifications

  1. Comment: Line 438: Ankin et al. (42) in their study on ice cream quality depending…inulin, noted….

Response: Thank you for your suggestion. We have made corresponding modifications

  1. Comment: Line 448: remove sentence “Besides, the study by Balthazar et al. (48)”

Response: Thank you for your suggestion. We removed this sentence.

  1. Comment: Line 451: inulin (1) and it was shown…..

Response: Thank you for your suggestion. We have made corresponding modifications

  1. Comment: Line 464: addition of what kind of taste that was enhanced would be useful

Response: Thank you for your suggestion. We have made corresponding modifications

22.  Comment: Line 467: which parameter in table 7 indicated intensification of mango-passion fruit essence – is it the odour additives, additives taste, off taste, or?

Response: Thank you for your suggestion. We have made corresponding modifications. Additives taste in table 7 indicated intensification of mango-passion fruit

  1.  Comment: Line 477: ice cream sample prepared with…

Response: Thank you for your suggestion. We have made corresponding modifications
